# CLIP-Driven Universal Model for Partially Labeled Organ and Pan-cancer Segmentation

Jie Liu[1], Alan Yuille[2], Yucheng Tang[3,*], and Zongwei Zhou[2,*]

[1] City University of Hong Kong
[2] Johns Hopkins University
[3] NVIDIA
yuchengt@nvidia.com,zzhou82@jh.edu

**Abstract.** Automatic multi-organ segmentation in medical image analysis is a crucial task with various applications in computer-aided diagnosis and treatment. Convolutional neural networks (CNNs) have shown success in segmenting abdominal organs in CT images, but challenges arise due to complex morphology, low tissue contrast, and limited fully labeled datasets. Learning from partially labeled datasets has emerged as a promising solution. However, assembling partially annotated datasets presents formidable challenges, including background inconsistency and label orthogonality. To address these challenges, this study introduces the Universal Model, which incorporates text embedding and a masked back-propagation mechanism with binary segmentation masks. A revised label taxonomy is maintained, and binary segmentation masks are generated for each class during image pre-processing. The CLIP-based label encoding enhances the anatomical structure of the universal model's feature embedding, and loss is only computed for classes with available labels.

**Keywords:** Partial Label · Organ Segmentation · Universal Model.

## 1 Introduction

Automatic multi-organ segmentation is a crucial task in medical image analysis, with applications in computer-aided diagnosis and treatment [16]. Deep learning techniques, particularly convolutional neural networks (CNNs), have been successfully applied to this task. However, segmenting abdominal organs in CT images presents challenges due to their complex morphology, low tissue contrast, and the scarcity of fully labeled datasets [17]. Learning from partially labeled datasets has emerged as a promising solution to address the limitations of fully labeled data.

Formidable challenges exist in assembling partially annotated datasets. First, Background inconsistency. For example, the pancreas may be marked as the background in one volume, but it should have been marked as the foreground. Second, label orthogonality. Most segmentation methods, trained with one-hot labels [32], ignore the semantic relationship between classes. Given one-hot labels

of liver [1,0,0], liver tumor [0,1,0], and pancreas [0,0,1], there is no semantic difference between liver↔liver tumor and liver↔pancreas. A possible solution is few-hot labels [24], with which, the liver, liver tumor, and pancreas can be encoded as [1,0,0], [1,1,0], and [0,0,1]. Although few-hot labels could indicate that liver tumors are part of the liver, the relationship between organs remains orthogonal.

To address above mentioned challenged, *Universal Model* incorporates text embedding and adopts masked back-propagation mechanism with binary segmentation mask. Specifically, we maintain a revised label taxonomy derived from a collection of public datasets and generate a binary segmentation mask for each class during image pre-processing. For architecture design, we draw inspiration from Guo *et. al.* [8] and replaced one- or few-hot labels with the text embedding generated by the pre-trained text encoder from CLIP[4]. This CLIP-based label encoding enhances the anatomical structure of universal model feature embedding. At last, we only compute loss for the classes with available labels.

## 2    Method

### 2.1    Background

***Problem definition.*** Let $M$ and $N$ be the total number of datasets to combine and data points in the combination of the datasets, respectively. Given a dataset $\mathcal{D} = \{(\boldsymbol{X}_1, \boldsymbol{Y}_1), (\boldsymbol{X}_2, \boldsymbol{Y}_2), ..., (\boldsymbol{X}_N, \boldsymbol{Y}_N)\}$, there are a total of $K$ unique classes. For $\forall n \in [1, N]$, if the presence of $\forall k \in [1, K]$ classes in $\boldsymbol{X}_i$ is annotated in $\boldsymbol{Y}_i$, $\mathcal{D}$ is a *fully labeled* dataset; otherwise, $\mathcal{D}$ is a *partially labeled* dataset.

***Previous solutions.*** Two groups of solutions were proposed to address the partial label problem. Given a data point $\boldsymbol{X}_n, n \in [1, N]$, the objective is to train a model $\mathcal{F}(\cdot)$ using the assembly dataset $\mathcal{D}_A = \{\mathcal{D}_1, \mathcal{D}_2, ..., \mathcal{D}_M\}$, and the model can predict all $K$ classes, if presented in $\boldsymbol{X}_n$.

- *Solution #1* [5,24,29,24,33,3,12,26] aims to solve $\mathcal{F}_\theta(\boldsymbol{X}_n) = \boldsymbol{P}_n^k, n \in [1, N], k \in [1, K]$, where the prediction $\boldsymbol{P}_n$ is one-hot encoding with length $k$.
- *Solution #2* [32,14,34] aims to solve $\mathcal{F}_\theta(\boldsymbol{X}_n, \boldsymbol{w}_k) = \boldsymbol{P}_n, n \in [1, N], k \in [1, K]$, where $\boldsymbol{w}_k$ is an one-hot vector to indicate which class to be predicted.

According to Zhang et al. [32], both solutions have similar segmentation performance, but #2 is computationally more efficient. However, both solutions rely on one-hot labels, sharing two limitations. First, they ignore the semantic and anatomical relationship between organs and tumors. Second, they are inappropriate for segmenting various subtypes of tumors. To address these limitations, we modify $\boldsymbol{w}_k$ in Solution #2 to CLIP embedding and introduce in-depth in the following sections.

---

[4] CLIP (Contrastive Language–Image Pre-training) was pre-trained on 400 million image-text pairs (some are medical images and text [2]), exploiting the semantic relationship between images and language.

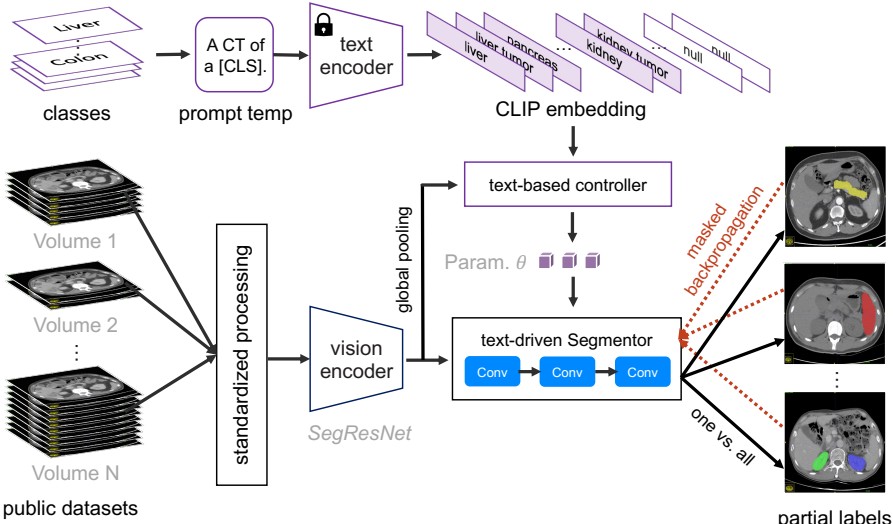

**Fig. 1.  Overview.** To deal with partial labels, Universal Model consists of a text branch and a vision branch (§2.2).

## 2.2   Universal Model

The overall framework of CLIP-Driven Universal Model (see Figure 1) has a text branch and a vision branch. The text branch first generates the CLIP embedding for each organ and tumor using an appropriate medical prompting, and then the vision branch takes both CT scans and CLIP embedding to predict the segmentation mask[5].

***Text branch.*** Let $w_k$ be the CLIP embedding of the $k$-th class, produced by the pre-trained text encoder in CLIP and a medical prompt (e.g., , "a computerized tomography of a [CLS]", where [CLS] is a concrete class name). We first concatenate the CLIP embedding ($w_k$) and the global image feature ($f$) and then input it to a multi-layer perceptron (MLP), namely *text-based controller* [27], to generate parameters ($\theta_k$), i.e., , $\theta_k = \mathrm{MLP}(w_k \oplus f)$, where $\oplus$ is the concatenation. Although CLIP embedding significantly outperforms one-hot labels [32], we mark that the choice of medical prompt template is critical.

---

[5] Our framework design is conceptually similar to *Segment Anything Model (SAM)* [15], which is a concurrent study of ours in computer vision. By leveraging CLIP embedding as a prompt within our Universal Model, we are able to generate highly accurate masks for organs and tumors of interest, as opposed to producing masks for arbitrary objects.

***Vision branch.*** We pre-process CT scans using isotropic spacing and uniformed intensity scale to reduce the domain gap among various volumes[6]. The standardized and normalized CT scans are then processed by the vision encoder. To facilitate the inference speed, we employ the light weight network SegResNet as backbone. Let $\boldsymbol{F}$ be the image features extracted by the vision encoder. To process $\boldsymbol{F}$, we use three sequential convolutional layers with $1 \times 1 \times 1$ kernels, namely *text-driven segmentor*. The first two layers have 8 channels, and the last one has 1 channel, corresponding to the class of $[\text{CLS}]_k$. The prediction for the class $[\text{CLS}]_k$ is computed as $\boldsymbol{P}_k = \text{Sigmoid}\left(((\boldsymbol{F} * \boldsymbol{\theta}_{k_1}) * \boldsymbol{\theta}_{k_2}) * \boldsymbol{\theta}_{k_3}\right)$, where $\boldsymbol{\theta}_k = \{\boldsymbol{\theta}_{k_1}, \boldsymbol{\theta}_{k_2}, \boldsymbol{\theta}_{k_3}\}$ are computed in the text branch, and $*$ represents the convolution. For each class $[\text{CLS}]_k$, we generate the prediction $\boldsymbol{P}_k \in \mathbb{R}^{1 \times D \times W \times H}$ representing the foreground of each class in *one vs. all* manner (i.e., , Sigmoid instead of Softmax).

***Masked back-propagation.*** To address the label inconsistency problem, we proposed the masked back-propagation technique. The BCE loss function is utilized for supervision. We masked the loss terms of these classes that are not contained in $\boldsymbol{Y}$ and only back-propagate the accurate supervision to update the whole framework. The masked back-propagation addresses the label inconsistency in the partial label problem. Specifically, partially labeled datasets annotate some other organs as background, leading to the disability of existing training schemes (Solution #1).

Moreover, we utilize the pseudo-label for unlabeled data to training the model to imporve the performance. During the inference, we conduct connected component analysis for different organs.

## 3   Experiments

### 3.1   Dataset and evaluation measures

The FLARE 2023 challenge is an extension of the FLARE 2021-2022 [19][20], aiming to aim to promote the development of foundation models in abdominal disease analysis. The segmentation targets cover 13 organs and various abdominal lesions. The training dataset is curated from more than 30 medical centers under the license permission, including TCIA [4], LiTS [1], MSD [25], KiTS [10,11], autoPET [7,6], TotalSegmentator [28], and AbdomenCT-1K [21]. The training set includes 4000 abdomen CT scans where 2200 CT scans with partial labels and 1800 CT scans without labels. The validation and testing sets include 100 and 400 CT scans, respectively, which cover various abdominal cancer types, such as liver cancer, kidney cancer, pancreas cancer, colon cancer, gastric cancer, and so on. The organ annotation process used ITK-SNAP [31], nnU-Net [13], and MedSAM [18].

---

[6] A standardized and normalized CT pre-processing is important when combining multiple datasets. Substantial differences in CT scans can occur in image quality and technical display, originating from different acquisition parameters, reconstruction kernels, contrast enhancements, intensity variation, and so on [22,30,9].

The evaluation metrics encompass two accuracy measures—Dice Similarity Coefficient (DSC) and Normalized Surface Dice (NSD)—alongside two efficiency measures—running time and area under the GPU memory-time curve. These metrics collectively contribute to the ranking computation. Furthermore, the running time and GPU memory consumption are considered within tolerances of 15 seconds and 4 GB, respectively.

**Table 1.** Development environments and requirements.

| | |
|---|---|
| System | Ubuntu 18.04.5 LTS |
| CPU | Intel(R) Core(R) Gold 5317 CPU@3.00GHz |
| RAM | 16×4GB; 2.67MT/s |
| GPU (number and type) | Four NVIDIA 4090 24G |
| CUDA version | 11.8 |
| Programming language | Python 3.20 |
| Deep learning framework | torch 2.0, torchvision 0.2.2, monai 1.1.0 |

**Table 2.** Training protocols.

| | |
|---|---|
| Network initialization | He Initialization |
| Batch size | 4 |
| Patch size | 32×192×192 |
| Total epochs | 300 |
| Optimizer | NovoGrad |
| Initial learning rate (lr) | 1e-4 |
| Lr decay schedule | Cosine Annealing |
| Training time | 72.5 hours |
| Loss function | BCE and Dice Loss |
| Number of model parameters | 4.69M[7] |
| Number of flops | 81.93G[8] |
| $CO_2$eq | 1.38 Kg[9] |

### 3.2  Implementation details

**Environment settings** The development environments and requirements are presented in Table 1. The system is running Ubuntu 18.04.5 LTS as the operating system. The CPU in use is an Intel(R) Core(R) Gold 5317 CPU with a clock speed of 3.00GHz. The system has a total of 64GB RAM, divided into 16 modules of 4GB each, operating at a speed of 2.67MT/s. The system is equipped with four NVIDIA 4090 24G GPUs. The CUDA version installed on the system is 11.8. The

programming language used for development is Python 3.20. The deep learning framework employed includes torch 2.0, torchvision 0.2.2, and monai 1.1.0. These specifications provide insight into the hardware and software setup used for the development of a specific project or application.

**Training protocols** The training protocols are presented in Table 1. The network is initialized using He Initialization, and training is performed with a batch size of 4. The input patches during training have a size of 32x192x192. The training process runs for a total of 300 epochs using the NovoGrad optimizer with an initial learning rate of 1e-4. The learning rate decay schedule follows the Cosine Annealing method. The entire training process takes approximately 72.5 hours. The loss function used combines Binary Cross Entropy (BCE) loss and Dice loss. The network has a total of 4.69 million parameters and performs 81.93 billion floating-point operations (flops). The training process results in approximately 1.38 kg of $CO_2eq$ emissions.

**Table 3.** Quantitative evaluation results.

| Target | Public Validation | | Online Validation | | Testing | |
|---|---|---|---|---|---|---|
| | DSC(%) | NSD(%) | DSC(%) | NSD(%) | DSC(%) | NSD (%) |
| Liver | $97.77 \pm 0.01$ | $98.64 \pm 0.01$ | 97.65 | 98.15 | | |
| Right Kidney | $90.67 \pm 3.71$ | $90.82 \pm 4.00$ | 87.32 | 87.33 | | |
| Spleen | $96.58 \pm 0.03$ | $98.85 \pm 0.13$ | 97.08 | 98.87 | | |
| Pancreas | $81.66 \pm 3.10$ | $92.32 \pm 3.72$ | 82.48 | 93.75 | | |
| Aorta | $84.52 \pm 1.44$ | $87.53 \pm 1.49$ | 84.95 | 87.60 | | |
| Inferior vena cava | $87.83 \pm 0.52$ | $89.74 \pm 0.65$ | 87.76 | 89.60 | | |
| Right adrenal gland | $77.80 \pm 2.80$ | $91.77 \pm 3.65$ | 78.11 | 91.54 | | |
| Left adrenal gland | $76.65 \pm 1.75$ | $90.38 \pm 2.03$ | 75.62 | 87.88 | | |
| Gallbladder | $82.14 \pm 6.68$ | $83.33 \pm 7.52$ | 78.51 | 79.60 | | |
| Esophagus | $76.79 \pm 3.07$ | $87.15 \pm 3.40$ | 78.00 | 88.41 | | |
| Stomach | $92.88 \pm 0.16$ | $95.63 \pm 0.25$ | 91.92 | 94.40 | | |
| Duodenum | $77.30 \pm 1.10$ | $90.49 \pm 0.59$ | 77.07 | 89.23 | | |
| Left kidney | $91.00 \pm 3.49$ | $92.27 \pm 3.08$ | 88.19 | 87.89 | | |
| Tumor | $50.83 \pm 11.26$ | $41.00 \pm 8.95$ | 40.05 | 31.04 | | |
| Average | $85.66 \pm 2.14$ | $91.46 \pm 2.35$ | 84.97 | 90.35 | | |

## 4   Results and discussion

### 4.1   Quantitative and Qualitative results on validation set

Table 3 report the Dice and NSD scores of organs and tumors on the validation set. The targets include liver, right kidney, spleen, pancreas, aorta, inferior vena cava, right adrenal gland, left adrenal gland, gallbladder, esophagus, stomach, duodenum, left kidney, and tumor. The evaluation is performed on three sets: public validation, online validation, and testing. The segmentation performance

of the liver, as indicated by high DSC and NSD scores across all validation sets, demonstrates accurate segmentation. While the right kidney achieves slightly lower scores compared to the left kidney, both kidneys still exhibit reasonable segmentation performance. Similarly, the spleen segmentation showcases high DSC and NSD scores across all validation sets, indicating accurate segmentation. Although the pancreas segmentation yields lower scores compared to other organs, it still attains a reasonable level of segmentation accuracy. The segmentation performance for the aorta and inferior vena cava remains relatively consistent across the validation sets. On the other hand, the adrenal glands, gallbladder, esophagus, stomach, and duodenum exhibit varying levels of segmentation performance, with some structures achieving higher scores than others. Notably, the tumor segmentation demonstrates the lowest scores among all targets, highlighting it as the most challenging segmentation task. Furethermore, we show four example results in the validation set in Figure 2.

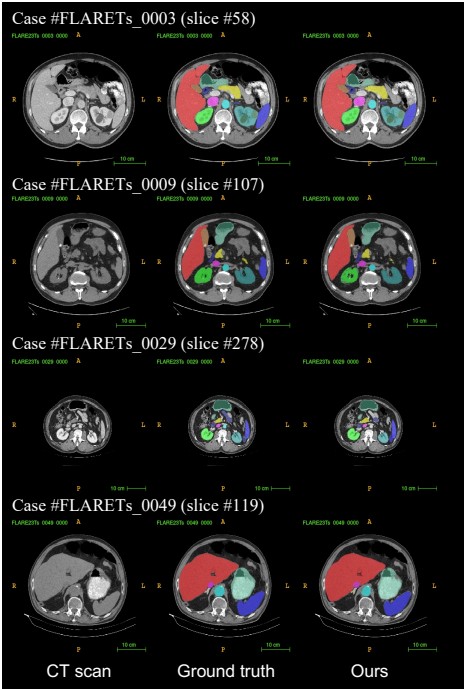

**Fig. 2.** Qualitative results on validation set.

### 4.2   Segmentation efficiency results on validation set

Table 4 presents a quantitative evaluation of segmentation efficiency in terms of running time and GPU memory consumption. The running time varies across different cases, ranging from 30.06 seconds to 91.43 seconds. As the image size

increases, the running time tends to increase as well. For example, the cases with larger image sizes, such as Case ID 0048 and 0029, have higher running times compared to the cases with smaller image sizes, such as Case ID 0001 and 0051. The table provides insights into the GPU memory usage during segmentation. The maximum GPU memory usage ranges from 2574 MB to 3240 MB, while the total GPU memory usage varies from 34,352 MB to 187,594 MB. Similar to the running time, the memory consumption tends to increase with larger image sizes.

**Table 4.** Quantitative evaluation of segmentation efficiency in terms of the running them and GPU memory consumption.

| Case ID | Image Size | Running Time (s) | Max GPU (MB) | Total GPU (MB) |
| --- | --- | --- | --- | --- |
| 0001 | (512, 512, 55) | 33.49 | 3198 | 34352 |
| 0051 | (512, 512, 100) | 30.06 | 3226 | 52476 |
| 0017 | (512, 512, 150) | 46.09 | 2938 | 72434 |
| 0019 | (512, 512, 215) | 45.73 | 2574 | 86161 |
| 0099 | (512, 512, 334) | 65.02 | 2928 | 130350 |
| 0063 | (512, 512, 448) | 89.16 | 3222 | 183466 |
| 0048 | (512, 512, 499) | 91.43 | 3218 | 187594 |
| 0029 | (512, 512, 554) | 69.74 | 3240 | 137061 |

### 4.3   Results on final testing set

This is a placeholder. We will send you the testing results during MICCAI (2023.10.8).

## 5   Conclusion

In conclusion, the development of the Universal Model for automatic multi-organ segmentation in CT images addresses the challenges posed by partially labeled datasets. By incorporating text embedding and a masked back-propagation mechanism, the model enhances the anatomical structure representation and overcomes issues of background inconsistency and label orthogonality. The use of binary segmentation masks generated during image pre-processing and the adoption of CLIP-based label encoding contribute to improved segmentation accuracy. This approach enables the model to effectively learn from partially labeled datasets, which are more readily available than fully labeled datasets. The Universal Model represents a promising solution for accurate and efficient multi-organ segmentation, with potential applications in computer-aided diagnosis and treatment planning in medical imaging. Further research and validation on diverse datasets are warranted to assess the generalizability and robustness of the Universal Model in real-world clinical settings.

**Acknowledgements** The authors of this paper declare that the segmentation method they implemented for participation in the FLARE 2023 challenge has not used any pre-trained models nor additional datasets other than those provided by the organizers. The proposed solution is fully automatic without any manual intervention. We thank all the data owners for making the CT scans publicly available and CodaLab [23] for hosting the challenge platform.

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

**Table 5.** Checklist Table. Please fill out this checklist table in the answer column.

| Requirements | Answer |
|---|---|
| A meaningful title | Yes |
| The number of authors ($\leq 6$) | 6 |
| Author affiliations, Email, and ORCID | Yes/No |
| Corresponding author is marked | Yes/No |
| Validation scores are presented in the abstract | Yes |
| Introduction includes at least three parts: background, related work, and motivation | Yes |
| A pipeline/network figure is provided | 3 |
| Pre-processing | 4 |
| Strategies to use the partial label | 4 |
| Strategies to use the unlabeled images. | 4 |
| Strategies to improve model inference | 4 |
| Post-processing | 4 |
| Dataset and evaluation metric section is presented | 4 |
| Environment setting table is provided | 5 |
| Training protocol table is provided | 5 |
| Ablation study | Page number |
| Efficiency evaluation results are provided | 6 |
| Visualized segmentation example is provided | 7 |
| Limitation and future work are presented | Yes |
| Reference format is consistent. | Yes |