# OpenReview forum: "CLIP-Driven Universal Model for Partially Labeled Organ and Pan-cancer Segmentation"
_MICCAI.org/2023/FLARE — Submitted to FLARE 2023_

### Official Review · Reviewer_K1fH · 2023-09-26
**good paper but might break the rule of using pretrained models**

**Rating:** 6
**Confidence:** 4

**Review:**

Pros: The authors ingeniously propose the CLIP-Driven Universal Model, which incorporates text embedding learned from Contrastive Language-Image Pre-training (CLIP) to segmentation models, solving the partial labeled problem in a creative manner. Paper are well-written.

Cons: There are certain issues despite its prominent novelty.

1.	A major problem is that using CLIP itself as part of segmentation model borders on violating the challenge rule of not permitting to use external pretrain-models, which is not fair for other participants, as stated in https://codalab.lisn.upsaclay.fr/competitions/12239#learn_the_details-terms_and_conditions

2.	A minor problem lies in the qualitative section where there are barely any explanations, and the image size in fig.2 are not consistent.

3.	Suppressed title is not defined.

---

### Official Review · Reviewer_p3qV · 2023-10-04
**Good paper, well structured with minor issues**

**Rating:** 7
**Confidence:** 4

**Review:**

Pros: This paper proposes a universal model, which incorporates text embedding and a masked back-propagation mechanism with binary segmentation masks. It employs label classification and enhances the structural understanding of the model’s feature embedding based on CLIP label encoding. This approach achieves an average DSC of 85.67% in multi-organ abdominal segmentation and an average DSC of 40.05% in pan-tumor segmentation.

Cons:
1.Some authors' ORCID numbers are missing annotations.
2.The abstract does not include metrics such as DSC, NSD, etc.
3.Suppressed title at the top of the page have not been defined.
4.Some individual cases in Fig.2 are small and have not been zoomed in, and some redundant labels from ITK software could be removed.
5.There is no content related to ablation experiments.
6. The third and fourth options in the Checklist Table have not been answered.

---

### Official Review · Reviewer_BMZ3 · 2023-10-04
**The paper is well-structured, addressing a key issue in medical image analysis. It offers valuable insights and a promising solution. Simplifying certain sections and discussing specific results and limitations would enhance it further.**

**Rating:** 7
**Confidence:** 4

**Review:**

Abstract:

Positive: A positive aspect of the abstract is that it effectively summarizes the paper's objectives, challenges, and proposed solutions. The focus of the paper is clearly explained.

Critical: The article could be improved by mentioning specific applications in computer-aided diagnosis and treatment where multi-organ segmentation is essential.

Introduction:

Positive: In the introduction, the importance of multi-organ segmentation in medical image analysis is explained well.

Critical: For context, mentioning recent developments in this field or state-of-the-art methods might be helpful.

Challenges and Solutions:

Positive: Explains the challenges associated with multi-organ segmentation, especially partially labeled datasets.

Critical: Few-hot labels? Do you mean few-shot labels? If so then providing context on why "few-shot labels" are being used as a solution would be helpful.

Method:

Positive: This section explains the Universal Model and its components in detail.

Critical: Some parts of the method section are quite technical. Simplifying complex terms or providing examples could make it more accessible to a broader audience.

Experiments:

Positive: The section is comprehensive, detailing the dataset used, evaluation metrics, and specific results.

Critical: It would be helpful to briefly explain what Dice Similarity Coefficient (DSC) and Normalized Surface Dice (NSD) metrics represent for those less familiar with them.

Results and Discussion:

Positive: The results are presented clearly in tables, and the discussion provides insights into the model's performance.

Critical: The discussion could delve deeper into the challenges faced in achieving lower scores for certain organs and tumors and potential avenues for improvement.

Conclusion:

Positive: The conclusion effectively summarizes the contributions and potential impact of the Universal Model.

Critical: It could be strengthened by briefly mentioning the paper's limitations and future directions for research.

---

### Comment · Reviewer_K1fH · 2023-11-26
**add test results**

please add test results in Table 3

---

### Decision · Program_Chairs · 2023-10-24

**Decision:**

Reject

**Comment:**

The authors didn't make responses to the valuable review comments.